# Boundary complexity of cortical and subcortical areas predicts deep brain stimulation outcomes in Parkinson's disease

While deep brain stimulation (DBS) remains an effective therapy for Parkinson's disease (PD), sources of variance in patient outcomes are still not fully understood, underscoring a need for better prognostic criteria. Here, we leveraged routinely collected T1-weighted (T1-w) magnetic resonance imaging (MRI) data to derive patient-specific measures of brain structure and evaluate their usefulness in predicting changes in PD medications in response to DBS. Preoperative T1-w MRI data from 231 patients with PD were used to extract regional measures of fractal dimension (FD), sensitive to the structural complexities of cortical and subcortical brain. FD was validated as a biomarker of PD progression through comparison of patients with PD and healthy controls (HCs). This analysis revealed significant group differences in FD across nine brain regions, including frontal, occipital, insular, and basal ganglia areas, which supports its utility as a marker of PD. We evaluated the impact of adding imaging features (FD) to a clinical model that included demographics and clinical parameters (age, sex, total number and location of DBS electrodes), and preoperative motor response to levodopa. This model aimed to explain variance and predict changes in medication following DBS. Regression analysis revealed that inclusion of the FD of distributed brain areas correlated with post-DBS reductions in medication burden, explaining an additional 13.6% of outcome variance ($R^2 = 0.388$) compared to clinical features alone ($R^2 = 0.252$). Hypergraph-based classification learning tasks achieved an area under the receiver operating characteristic curve of 0.64 when predicting with clinical features alone, versus 0.76 when combining clinical and imaging features. These findings demonstrate that PD effects on brain morphology linked to disease progression influence DBS outcomes. The work also highlights FD as a potentially useful imaging biomarker to enhance DBS candidate selection criteria for optimized treatment planning.

Parkinson's disease (PD) is a neurodegenerative disorder characterized by motor symptoms, including tremor, rigidity, bradykinesia, and postural instability, as well as non-motor symptoms, such as cognitive impairments and mood disturbances[1,2]. PD is marked by a pronounced loss of dopaminergic neurons in the brain, particularly in the substantia nigra[3]. Currently, PD affects millions of individuals worldwide, with higher incidence rates observed in older adults. As life expectancy continues to rise and

✉e-mail: melanie.morrison@ucsf.edu

the population ages, the burden of PD is expected to escalate further[4].

While there is no cure for PD, various options are available to manage symptoms and improve patients' quality of life. Medication remains the cornerstone of treatment, aiming to replenish dopamine levels or enhance its effectiveness[5]. However, in cases where medication can no longer control symptoms adequately, an adjunct therapy like deep brain stimulation (DBS) may be considered[6]. DBS is a neuromodulation device-based therapy that involves the surgical implantation of electrodes into targeted brain regions, typically the subthalamic nucleus (STN) and the globus pallidus interna (GPi), to deliver controlled electrical impulses that modulate abnormal neuronal activity and alleviate motor symptoms[7]. As the disease progresses, fluctuations in motor control with medication management alone are often a factor in determining the suitability of deep DBS as a therapy for a patient. DBS is useful in improving motor fluctuations, reducing medication-related side effects, including dyskinesia, and has been shown to improve patients' quality of life. Consequently, the use of DBS in PD is on the rise, as is the number of candidates due to the increasing prevalence of PD[8,9].

Despite its efficacy in alleviating PD motor symptoms in many patients, outcomes following DBS exhibit significant variability, with instances of notable improvement juxtaposed with cases displaying limited response or even worsening of symptoms[10–12]. Current selection criteria attempt to predict these outcomes based on the responsiveness of patients' motor symptoms to dopaminergic medications[13,14]. However, medication responsiveness has been shown to have limited predictive accuracy[15–18], presenting the need for more accurate and objective prognostic markers that can reliably identify the most ideal DBS candidates.

Magnetic resonance imaging (MRI) is integral to DBS implementation and thus a convenient tool for extracting quantitative biomarkers and predicting individual outcomes. Advanced MRI techniques probing brain function, tissue microstructure, and iron content via tissue magnetic susceptibility have previously been shown to be related to DBS outcomes[19–25], though these sequences are not always routinely collected as a part of routine clinical work-up. In contrast, T1-weighted (T1-w) anatomical MRI is universally collected for screening and treatment planning for every DBS patient. Despite its ubiquitous collection, researchers have yet to derive robust biomarkers from T1-w images, though leveraging their widespread availability and routine acquisition could offer valuable insights into individual treatment outcomes that are easily translatable to clinical practice.

Here, we focus on extracting regional brain measures of fractal dimension (FD) from T1-w images as potential complementary features to other clinical and imaging predictors to explain additional variance in PD DBS outcomes. FD quantifies the irregularity of a structure's surface[26,27], and in the context of PD and DBS, offers a unique perspective on the underlying structural changes associated with disease progression and treatment response. Specifically, FD can capture both cortical compensatory processes and subcortical degeneration, offering a more nuanced assessment of disease progression beyond conventional volumetric or thickness-based metrics that have previously been related to DBS motor outcomes[28]. Unlike cortical thickness and volumetric measures, which primarily capture atrophy, FD quantifies the complexity and self-similarity of structures[29,30]. This allows FD to detect microstructural irregularities and non-uniform remodeling patterns, which may precede detectable changes in traditional measures[31]. Previous work shows that FD can distinguish PD from normative brains[32,33], as well as detect subtle changes in neurodegeneration[34,35]. Based on these characteristics of FD and the brain's structure-function relationship, we hypothesize that baseline structural brain health informed by FD, influences and thus may be able to predict individual functional response to DBS.

The first part of this study validates FD as a biomarker of PD progression using open-source, multi-center data. We then leverage our large single-center dataset comprised of preoperative T1-w images from 231 individuals with PD and DBS and employ hypergraph techniques for data representation, variance modeling, and classification learning tasks, to evaluate the potential usefulness of FD as a biomarker for DBS candidate selection.

## Results
### Validation of FD as a PD biomarker
Validity of FD as a neurodegenerative biomarker was first investigated by comparing open-source, multi-center data from patients with PD ($n = 70$) and healthy controls (HCs; $n = 70$) and modeling FD against a proxy of disease severity in our large PD cohort ($n = 231$). A two-sided t-test with corrections for multiple comparisons revealed significant PD and HC group differences in FD ($p < 0.01$) for nine brain regions including: the left superior and medial frontal gyri, left supplementary motor area, left insular cortex, right inferior occipital area, bilateral paracentral lobules, right putamen, and right pallidum (Fig. 1A, B). Cortical regions showed an increase in FD for patients with PD compared to HCs, while subcortical regions demonstrated lower FD in the patient group. The classification power of selected FD features was tested using a hypergraph neural network (HGNN) learning model which achieved an area under the receiver operating characteristic curve (AUC) of 0.88 with sensitivity and specificity of 0.86 and 0.79, respectively, for distinguishing patients with PD from HCs (Fig. 1B, C). Regional FD showed significant correlations with disease severity in PD, inferred by patients' preoperative motor scores while off PD medications (Table 1). Specifically, significant positive correlations were found for FD of the left superior frontal gyrus, left supplementary motor area, left gyrus rectus, left parahippocampal region, right lingual gyrus, left postcentral gyrus, left inferior parietal lobule, and right superior temporal gyrus.

### Selected features to predict outcomes
For the main analysis, STN and GPi patients were combined, with the DBS target included as a covariate to maintain cohort size and to enhance generalizability and utility. Feature selection via LASSO regression on the training dataset ($n = 162$) yielded 20 significant predictors including: the DBS target (STN vs. GPi), hemispheres treated (bilateral vs. unilateral), age at MRI, percent improvement in preoperative motor scores from the OFF- to ON-medication state, and FD of 16 brain regions localized to the frontal lobe, Rolandic cortex, cingulate cortex, amygdala, occipital lobe, paracentral lobule, and caudate nucleus (Fig. 2A). A LASSO shrinkage factor of 0.03 was selected based on optimal classification performance of corresponding predictive features selected and tested across a range of shrinkage factors (Fig. 2C). Beyond the DBS target, FD of the right paracentral lobule, left middle occipital gyrus, and right olfactory area were the three most important features with the highest normalized ridge regression coefficients (Fig. 2B). Mapping selected FD predictors onto a normalized T1-w image visually confirmed their spatial distribution throughout the brain (Fig. 2D). Individual differences in average FD could also be visually appreciated directly from segmented tissue boundaries on subject-specific T1-w images (Fig. 2E).

### Variance in outcomes explained by FD
Ordinary least squares regression (OLS) was used to assess the variance in 6-month DBS outcomes (pre-to-post change in PD medications) explained by selected clinical and imaging features. Including selected FD features from the training dataset ($n = 162$) in a combined model with clinical features explained an additional 13.6% of the variance in patient outcomes ($R^2 = 0.388$) compared to a model of clinical features alone ($R^2 = 0.252$). An F-test comparing the two models

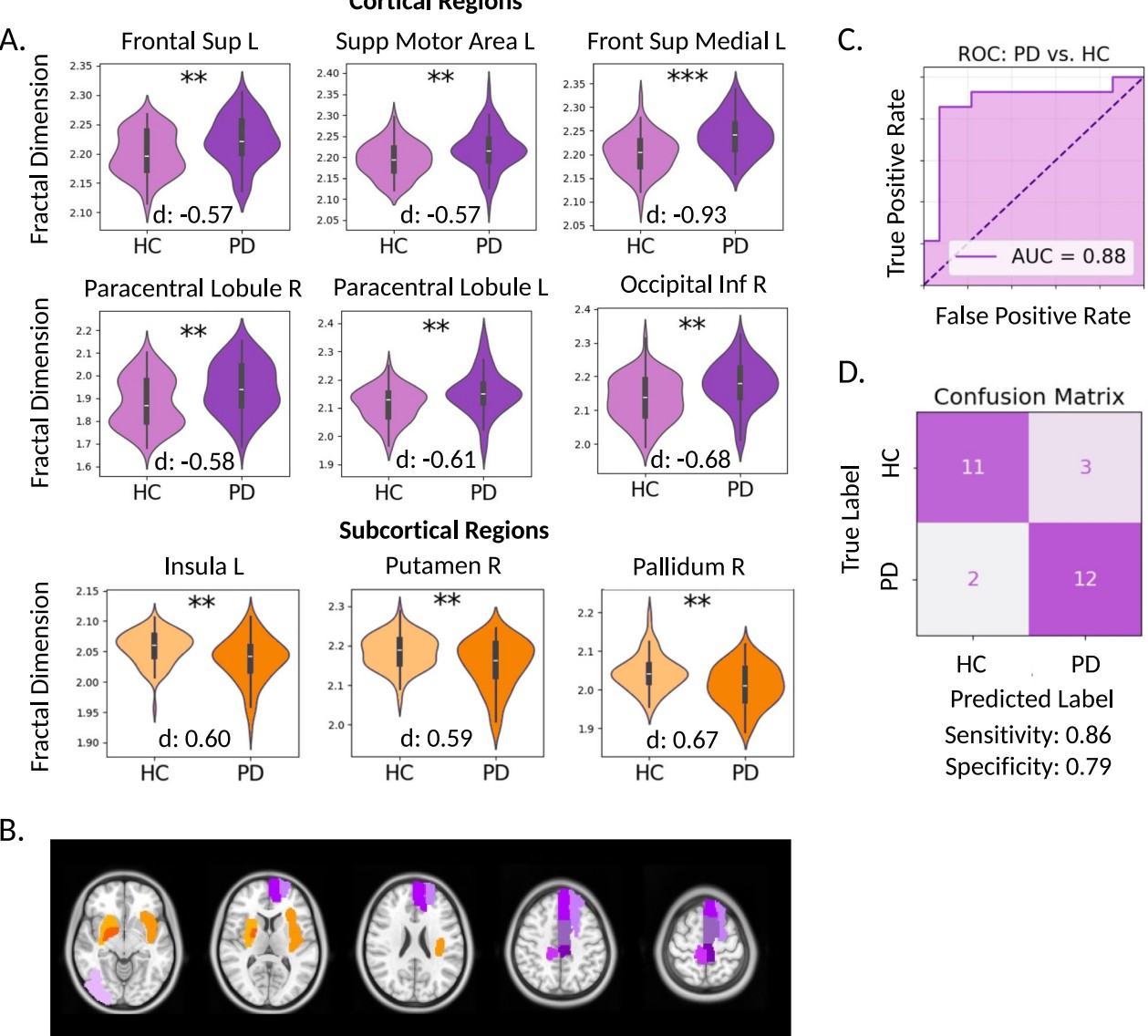

**Fig. 1 | Fractal dimension (FD) differentiates patients with Parkinson's disease (PD) from healthy controls (HCs). A** Nine brain regions showed significant FD differences between PD ($n = 70$) and HC ($n = 70$). Violin plots show the distribution of values for each group. Black bars represent median values; box limits indicate the interquartile range (25th to 75th percentile); whiskers extend to 1.5× the IQR. Statistical differences were assessed using two-sided t-tests with $p$-values adjusted for multiple comparisons using the Benjamini-Hochberg method. Cohen's d effect sizes quantify the magnitude of the differences. **B** Spatial distribution of the nine regions with significant FD differences. Purple regions indicate cortical areas where FD was higher in PD, while orange regions represent subcortical areas where FD was lower in PD. **C** Hypergraph neural network (HGNN) classification of PD vs. HC using the nine significant FD features. The model achieved an area under the receiver operating characteristic curve (AUC) of 0.88. **D** Confusion matrix for classification performance. The model achieved 86% sensitivity (correctly identifying PD) and 79% specificity (correctly identifying HC). Source data are provided as a Source Data file.

yielded an F-statistic of 2.76 ($p < 0.01$), confirming that the inclusion of FD significantly improved the model fit.

**Target-specific differences in FD predictions**

None of the selected FD predictors demonstrated a statistically significant interaction term ($p > 0.05$), indicating that the relationship between FD and ΔLEDD did not differ by DBS target. Nevertheless, the interaction plots reveal distinct slope patterns across regions (Fig. 3). In the left superior frontal gyrus, right inferior opercular frontal gyrus, right olfactory cortex, bilateral amygdala, left cuneus, right inferior parietal cortex, and left paracentral lobule, slopes for the two DBS targets were similar in both sign and magnitude, suggesting a comparable relationship between FD and ΔLEDD. Conversely, in the left inferior orbital frontal gyrus, left Rolandic operculum, left middle

cingulum, left superior occipital cortex, right paracentral lobule, left caudate, and right middle temporal pole, target-specific slopes had opposite signs, suggesting a potential target-dependent divergent effect of FD on ΔLEDD. Finally, while target-specific slopes for the left middle occipital gyrus were of the same sign, there was a pronounced difference in magnitude, with a steeper slope and thus stronger interaction effect observed for STN outcomes.

**Classification performance of FD**

An HGNN classification learning task was performed to distinguish patients whose daily medication dose decreased versus increased (or did not change) following DBS. Optimal performance of the combined model was achieved using a model hyperparameter of $k = 12$, which defines how many nodes each hyperedge contains in a k-uniform

**Table 1 | Relationships between regional FD and preoperative MDS-UPDRS III motor scores (OFF-medication state) were assessed using linear regression**

| Region | p | R | 95% CI |
|---|---|---|---|
| Frontal Sup Orb L | * | 0.12 | [0.009, 0.248] |
| Supp Motor Area L | * | 0.13 | [0.001, 0.258] |
| Rectus L | * | 0.12 | [−0.003, 0.249] |
| Parahippocampal L | * | 0.13 | [0.002, 0.259] |
| Lingual R | ** | 0.21 | [0.078, 0.328] |
| Postcentral L | * | 0.13 | [0.000, 0.255] |
| Parietal Inf L | ** | 0.16 | [0.032, 0.286] |
| Temporal Sup R | * | 0.13 | [0.005, 0.252] |

*$p < 0.05$, **$p < 0.01$.
Two-sided $p$-values are reported Table includes correlation coefficients ($r$), 95% confidence intervals, and adjusted $p$-values. $P$-values were corrected for multiple comparisons using the Benjamini–Hochberg method. Sample size: $n = 231$ patients.

hypergraph (Fig. 4A). Combining FD features with clinical features in a multi-feature hypergraph model trained and optimized on 196 patients and tested on 35, resulted in an AUC of 0.76 compared to 0.64 for a hypergraph model derived from clinical features alone (Fig. 4B). The inclusion of FD with clinical features increased the model's sensitivity by 15 percentage points (from 0.54 to 0.69) and specificity by 22 percentage points (from 0.56 to 0.78), enhancing the overall performance in predicting changes in levodopa equivalent daily dose (LEDD) (Fig. 4C).

## Discussion

This retrospective study investigated the added value of T1-w measures of cortical and subcortical complexity in predicting PD outcomes with DBS. Few prior studies have used T1-w FD to discriminate patients with PD from HCs and reported mixed findings. Kubera et al. reported no significant differences in FD between 22 patients with PD and 18 HCs[32]. Conversely, Li et al. investigated a larger sample of 60 patients and 56 HCs and identified significant reductions in FD for

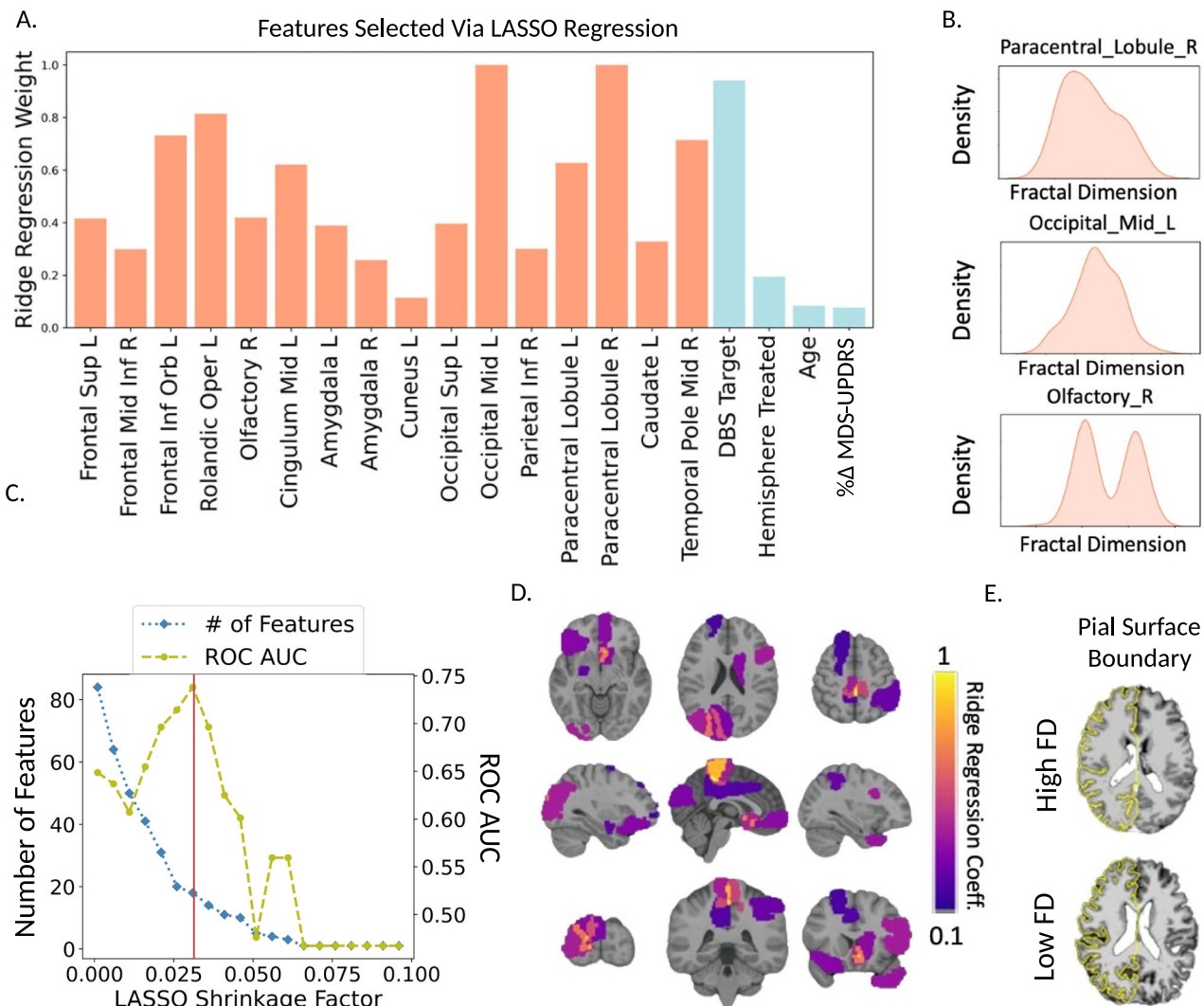

**Fig. 2 | Relevant imaging and clinical features selected via LASSO regression.**
**A** Sixteen regional fractal dimension (FD) features and four clinical features were selected. The DBS target was one of the most important predictors of medication change following DBS, along with FD of three brain areas (**B**): the right paracentral lobule, left middle occipital gyrus, and right olfactory area. ROC AUC=area under receiver operating characteristic curve. **C** The LASSO shrinkage factor impacted the number of selected features and corresponding classification performance. The

vertical red line denotes the optimal shrinkage factor used in the analysis.
**D** Regional FD features selected via LASSO regression were spatially distributed throughout the brain. Warmer colors on the heat map denote features with higher ridge regression coefficients and thus greater importance for predicting DBS outcomes. **E** Comparison of reconstructed pial surface boundaries overlaid on T1-weighted images of two patients with high and low average FD, revealed visual differences in cortical structure. Source data are provided as a Source Data file.

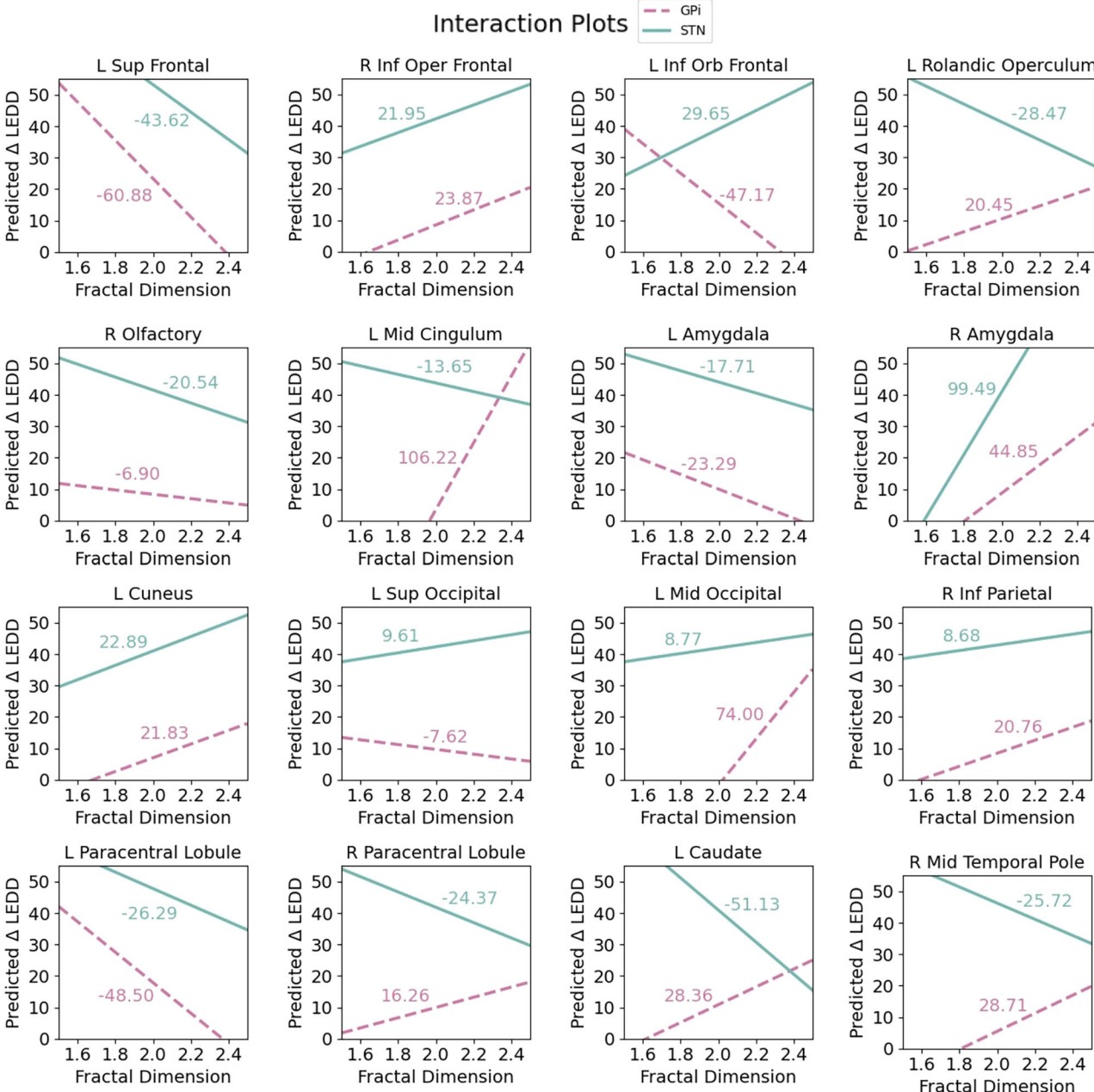

**Fig. 3 | Target-specific differences in fractal dimension (FD) predictions.** Interaction plots for the predicted pre-to-post-DBS change in Parkinson's medication (ΔLEDD) as a function of FD across selected brain regions, comparing globus pallidus internus (GPi) (dashed lines) and subthalamic nucleus (STN) (solid lines) targets. Slope values for each target are indicated on each plot. Source data are provided as a Source Data file.

patients, particularly in cortical motor areas, indicating that FD could capture important cortical changes associated with PD progression[33]. Our findings add to this small body of literature and echo other evidence of FD robustness across multiple acquisitions[36] by demonstrating that FD can effectively distinguish patients with PD from HCs using an unbiased classifier neural network model and open-source, multi-center data. The direction of FD group effects was region-specific, which we hypothesize could be reflective of distinct underlying neurobiological processes at the subcortical and cortical levels. Lower FD values in PD subcortical areas such as the basal ganglia could indicate structural degeneration typical of PD progression[37,38]. In comparison, higher FD values in cortical areas integral to motor, cognitive, and visuospatial functions affected in PD, may be due to the presence of inflammatory processes or cellular changes as hypothesized by one group which observed increased

gray matter FD in patients with multiple sclerosis, compared to HCs[39]. This phenomenon of abnormally increased cortical complexity has also been observed in the motor and temporal cortices of patients with PD in schizophrenia[40], PD with hyposmia[41], and pre-manifest Huntington's disease[42]; as well as in manic-depressive patients over three decades ago[43]. A recent study in PD furthermore highlighted the importance of cortical compensatory processes in determining clinical disease severity, in conjunction with basal ganglia degeneration[44], supporting a distinction between subcortical and cortical disease processes affecting structural integrity. This distinction between cortical and subcortical FD patterns may be particularly relevant for understanding heterogeneous responses to DBS, given that individual brain structure characteristics directly mediate network functions, ultimately influencing how one functionally and clinically responds to therapy.

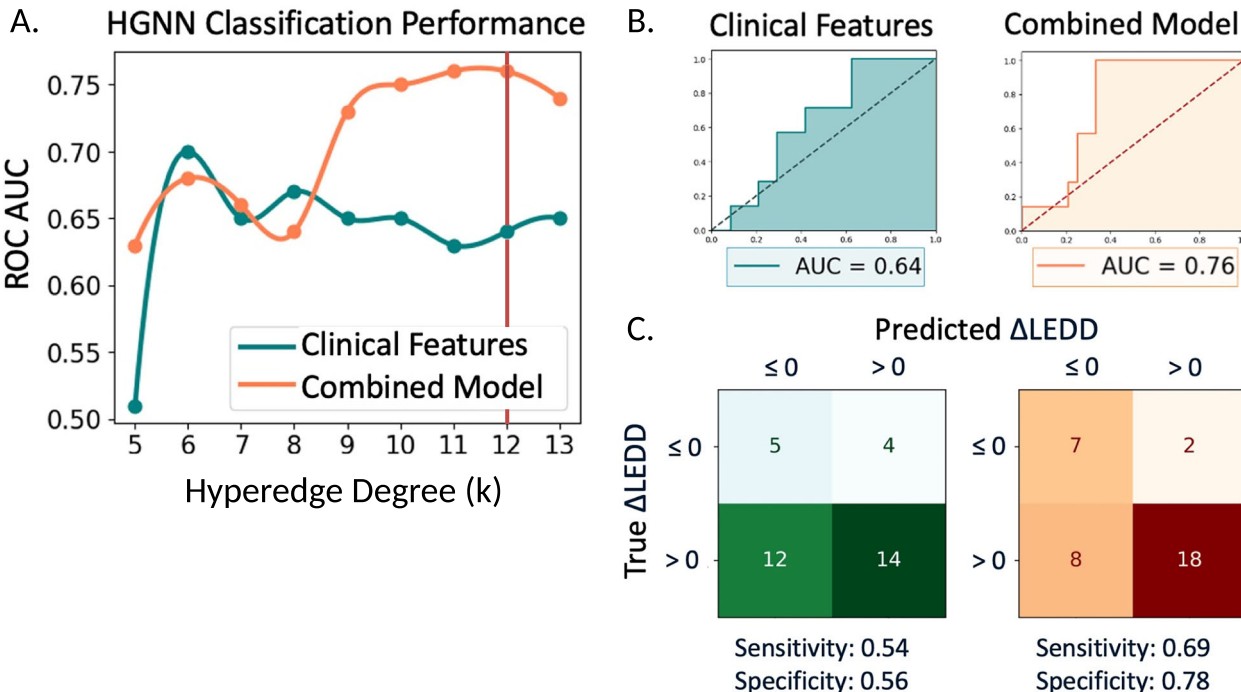

**Fig. 4 | Hypergraph neural network (HGNN) classification performance.**
**A** Optimal performance of the HGNN was achieved using a hypergraph topology k-value of 12 denoted by the vertical red line. **B** The combined model with imaging and clinical features achieved an area under the receiver operating characteristic curve (AUC) of 0.76, compared to 0.64 for the model with clinical features alone. **C** Classification of the test data ($n = 35$) using the combined model respectively increased sensitivity and specificity by 15% and 22%. Source data are provided as a Source Data file.

The fact that our group-level trends for cortical FD contradict observations by Li and colleagues despite a relatively similar sized cohort, suggests the presence of other sources of variability such as the cohort demographics or imaging methodology. For example, disease durations of our multi-site sub-cohorts were higher than the Li et al. cohort (3–4 years; range of approx. 1–10 years) at 9.4±4.9 years for our single-center data and 5.9±3.9 years for one of the two public datasets[45] (unreported for the other dataset). As the present study was not designed to fully resolve these inconsistencies, future prospective investigations are encouraged to refine existing biological interpretations of FD. Instead, the current work was focused on evaluating the biomarker potential of FD. The fact that FD could distinguish patients with PD from healthy controls with good predictive power, reinforced its specificity to PD characteristics and suitability for predicting PD outcomes with DBS.

Our within-patient trends also reinforced the use of FD as a biologically meaningful biomarker. Regional FD correlated with pre-operative motor symptom severity, measured using part III of the Movement Disorder Society Unified PD Rating Scale (MDS-UPDRS III) in the OFF-medication state[46]. We used the virtual version of the MDS-UPDRS III scores which did not include tests of rigidity or postural instability because of limited in person visits during the pandemic. The FD of frontal cortical areas showed consistent positive correlations with motor severity, further indicating increased cortical complexity with greater disease severity. Indeed, FD of frontal areas were also significant in distinguishing patients from HCs, with patients showing increased cortical complexity within these structures on average. Collectively, these results suggest that the structural integrity of the frontal cortex plays a significant role in objectively characterizing PD. As our subsequent results showed, this structural integrity of the frontal cortex was also predictive of patient outcomes, in line with one prior study which demonstrated in 31 patients that frontal cortical thickness could predict clinical improvement following STN-DBS[47].

This relationship between frontal cortical FD and outcomes may reflect broader network effects, given the frontal cortex's connectivity with motor-related regions such as the precuneus and basal ganglia, including the pallidum. The precuneus, for instance, is interconnected with the premotor area and is involved in the complex coordination of motor, cognitive, and emotional functions. Similarly, the pallidum, which serves as a target for DBS in alleviating motor symptoms, may be indirectly affected by changes in cortical structure and function. These observations suggest that structural and network-level changes in the frontal cortex and its connected regions contribute to the pathophysiology of PD and influence clinical outcomes. It is also worth noting that correlations between FD and motor severity were slightly stronger in the ON- versus OFF-medication state, which could reflect a better approximation of the functional disease state mediated by structural integrity.

LASSO and ridge regression analysis revealed key features impacting DBS outcomes. The selected DBS target explained the most variance in patient outcomes, consistent with longstanding evidence that clinical response to STN and GPi DBS can differ[48,49]. The relative importance of this feature was expected to be high and is directly related to our chosen outcome metric which approximated response to DBS based on pre-to-post changes in dopamine medications. STN stimulation is known for effectively reducing motor symptoms and medication needs, while GPi stimulation, often chosen for patients with prominent dyskinesias as well as greater risk for cognitive decline, typically results in less medication reduction[50]. While we combined STN and GPi groups in the main analysis to maintain generalizability, given the importance of this clinical feature, we also evaluated the relationship between FD and ΔLEDD for each DBS target, separately. The results suggest that while certain regional FD features were selected for their predictive utility in the main analysis, the effects of FD on ΔLEDD within each DBS target group may follow distinct patterns. However, due to the lack of statistical significance for the

interaction terms, these target-specific trends should be carefully interpreted, as they may reflect variability within the data rather than meaningful differences. Future work will verify the significance of these relationships as new data are added into the analysis and the study is expanded.

Other clinical features were also important, though less than regional FD features according to their ridge regression coefficient. A bilateral lead implantation was associated with more substantial symptom relief, consistent with clinical observations[51,52]. Age at the time of MRI was another impactful factor, with older patients potentially having different neural responses and adaptive capacities[53,54]. Finally, preoperative medication responsiveness, one of the gold standard clinical predictors of motor response to DBS, was also significantly related to outcomes. Although DBS acts through mechanisms distinct from pharmacological intervention[55], motor symptoms being levodopa responsive suggests that the network can be modified and that there are intact dopaminergic pathways. Our results highlight this value but also emphasize the added value of more objective predictors like FD.

Indeed, incorporating regional FD metrics into our model explained additional variance in DBS outcomes. Significant brain areas were distributed throughout the brain which is not surprising given the complex symptomatic nature of PD[56]. For example, important FD predictors localized to the frontal lobe, amygdala, cingulate gyrus, insula, and temporal pole suggesting that DBS outcomes may depend not only on the structural integrity of motor circuitry but also areas involved in cognitive and emotional processing. Evidence of patients with cognitive and/or psychiatric symptoms in PD experiencing less motor benefit and more cognitive side effects after DBS[57,58], certainly aligns with this hypothesis.

FD of the paracentral lobule was one of the most informative features, with a ridge regression coefficient comparable to the chosen DBS target, reinforcing the relationship between structural integrity of motor areas and response to DBS[25]. More specifically, given the role of the paracentral lobule in controlling lower extremity function, these results could potentially be indicative of a link between axial disability in PD and DBS outcomes. Indeed, research has shown that while STN-DBS may improve tremor and rigidity, its effects on axial symptoms, particularly gait and balance, are often less pronounced and can even worsen in some cases[59]. Another interesting finding worth noting is the predictive value of FD of the olfactory lobe. Olfactory dysfunction, particularly hyposmia, is a prevalent non-motor symptom in PD, with incidence rates ranging between 62% and 98%[60]. The double peak observed in the density plots for FD of the olfactory lobe (Fig. 2B) may reflect this variation in the severity of olfactory dysfunction among patients[41], reiterating the potential broader value of FD as a symptom-specific PD biomarker, though future dedicated studies are needed.

While variance analysis provides insights into factors driving DBS outcomes, practical application of imaging biomarkers requires predictive capability. Integrating FD metrics with clinical features in our HGNN model significantly improved predictive performance, with the combined model achieving higher performance metrics than the model trained on clinical features alone. Although predictive performance of the current model is moderate ($R^2 = 0.388$, AUC = 0.76), it represents an improvement over existing clinical measures and serves as a foundational step toward a more comprehensive predictive framework. We anticipate further enhancements through multimodal integration[19–24] and validation in multi-center datasets.

Clinically, DBS candidates are currently selected based on subjective predictors. A standardized approach to more objectively estimate the likelihood that an individual will respond well to the therapy has not yet been adopted. As the annual number of implantation surgeries continues to increase[8,9], there is a growing need for streamlined approaches to implementing DBS. FD can be computed in computationally inexpensive manner from standard T1-w MRI data that are readily accessible in all clinical DBS settings, making our approach feasible for widespread adoption. A key implication of our findings is that, by using MRI-derived biomarkers like FD, clinicians may eventually be able to predict which patients will benefit most from DBS without the need for traditional ON-/OFF-medication motor testing. This could streamline the decision-making process and reduce the burden on patients undergoing DBS candidacy evaluations. It would also allow clinicians to better target DBS resources toward patients most likely to benefit and lead to more democratized access to advanced DBS planning tools and potentially broadening the availability of DBS for a larger and more diverse population of patient with PD.

Finally, we would like to acknowledge that this study has several limitations. Automated registration and segmentation tools, while efficient, do not match the accuracy of manual segmentation. Also, due to retrospective data constraints, we used the pre-to-post change in dopaminergic medications as the primary outcome metric, which may not capture the full spectrum of motor and non-motor outcomes. For example, we did not consider the presence of dyskinesia or dopamine dependent depression which could influence neurologist strategies for modifying medication schedules following DBS. While medication outcomes are accessible at most centers and this broadens the applicability of our current models, to further enhance impact, we are working to extend our models to predict standardize clinical outcomes through prospective data collection. Other notable limitations include the use of virtual and thus partial preoperative motor scores as a proxy of disease severity; without rigidity and postural stability testing, the final scores used in this study may not accurately reflect patients' true total motor scores. Clinical data variability due to different sites, scanners, and protocols over thirteen years furthermore pose challenges by introducing differences in data quality. However, this variability also reflects real-world logistics of acquiring clinical data, even at a single center, and thus enhances model generalizability. Finally, while our model incorporates key clinical variables, other potentially relevant factors like psychiatric comorbidities and phenotypic variations—of which FD may be sensitive to[35]—were not available in a structured format for all patients. Surgical factors like lead location were also excluded, though intentionally, in line with our goal of developing a preoperative decision-support tool. Other ongoing work integrating lead and stimulation parameters to refine model predictions, will contribute to a more complete understanding of factors influencing DBS outcomes. Given these limitations, we encourage validation and refinement of our findings using our published data (see methods section for access), other datasets with standardized imaging protocols, and more accurate measures of outcome such as change in on and off medication MDS-UPDRS III and neurocognitive assessments after DBS. As we hope to expand this cohort to include multi-center data, we also welcome collaborations and data sharing efforts to further refine predictive modeling approaches for DBS outcomes in PD.

In summary, this work demonstrates the potential of T1-w MRI-derived FD as a valuable biomarker for predicting PD from HC and PD DBS outcomes. This research represents a significant step towards personalized medicine, enhancing the efficacy and accessibility of DBS therapy for a broader patient population.

## Methods
### Study population
Preoperative MRI data were retrospectively collected from 231 patients (mean age $65 \pm 9$ years; 33% female) who underwent DBS surgery at the University of California, San Francisco (UCSF) between 2010 and 2023. With approval from UCSF's institutional review board, all participants provided written informed consent to the use of their data for investigations of generic DBS outcomes. No compensation was provided for participation. A comprehensive list of patient demographics and clinical parameters is provided in Table 2. Sex was

**Table 2 | Demographics and treatment parameters of primary PD cohort**

| Clinical Feature | Total | ΔLEDD ≤ 0 | ΔLEDD > 0 |
|---|---|---|---|
| Subjects (n) | 231 | 60 | 171 |
| Male (n) | 155 | 40 | 115 |
| Female (n) | 76 | 20 | 56 |
| Mean Age (years) | 65 ± 9 | 66 ± 7 | 64 ± 10 |
| STN Target (n) | 96 | 6 | 90 |
| GPi Target (n) | 135 | 54 | 81 |
| Unilateral (n) | 44 | 15 | 29 |
| Bilateral (n) | 187 | 45 | 142 |
| Mean Motor Improvement (%)[a] | 53 ± 16 | 54 ± 15 | 52 ± 17 |
| Mean ΔLEDD (%)[b] | 23 ± 36 | −22 ± 22 | 39 ± 25 |

*LEDD* levodopa equivalent daily dose, *STN* subthalamic nucleus, *GPi* Globus Pallidus Interna
[a]Based on preoperative MDS-UPDRS III exam on- and off-medications.
[b]Based on patients' dosage regimen before versus after approx. 6 months of DBS.

self-reported by participants during clinical intake. No sex-specific analyses were conducted due to limited sample size and lack of a priori hypotheses. Aggregate sex breakdowns are provided in Table 2, and disaggregated sex information is included in the source data files. A majority of patients (80%) received bilateral DBS implants, with 58% of all subjects implanted in the GPi and the remaining 42% in the STN. Subjects were separated into two outcome groups according to changes in their LEDD following approximately 6 months of active DBS with optimized stimulation parameters and medication schedules. The first group included patients with a ΔLEDD *less* than or equal to 0 who experienced no change in medication or who increased their daily medication dose, while the second group consisted of patients with a ΔLEDD greater than zero who reduced their daily medication dose. A thorough description of this outcome metric is provided below.

### Outcome metric

A prospective study is underway to collect standardized motor and cognitive outcome data for this cohort. Thus, this retrospective analysis inferred outcomes from documented changes in patients' dopaminergic medications before versus after DBS surgery. Although ΔLEDD is not a direct measure of symptom response, it has previously been used as a surrogate measure to gauge clinical outcomes following DBS. Studies have shown that a reduction in LEDD generally correlates with positive motor outcomes, while an increase may indicate a suboptimal response to therapy[61,62].

To calculate LEDD for each patient, a comprehensive review of medical records was conducted to extract medication schedules from all available patient visits up to 12 months before and 12 months after DBS surgery. The type, dosage, and intake frequency of each dopaminergic medication were meticulously documented by three experienced researchers (D.S., J.M., S.D.). Daily medication dosages were converted to their levodopa equivalent using literature-reported conversion factors[63], accounting for bioavailability and pharmacokinetics and enabling the comparison and aggregation of medication doses across different regimens. The pre-to-post change in LEDD was calculated as

$$\Delta LEDD = 100\% \times \frac{LEDD_{pre} - LEDD_{post}}{LEDD_{pre}} \qquad (1)$$

with $LEDD_{pre}$ derived from the visit preceding DBS surgery and $LEDD_{post}$ from the visit nearest to 6 months of active DBS. This latter time point was selected based on data availability and the typical duration of DBS parameter optimization. Before incorporating ΔLEDD

in our models, individual and target-specific trends were explored, reinforcing target effects and revealing two divergent patterns in LEDD over time that informed clinical feature selection and subject classification labels (Fig. 5).

### Image acquisition and FD quantification

A flowchart illustrating key methodological steps is provided in Fig. 6A. Preoperative imaging was performed between 2010 and 2023 on 3 T MRI scanners of varying vendors and models with an 8- or 32- channel head coil. As this was a retrospective study, imaging protocols varied slightly across scans. Despite this natural variability, the acquisition parameters remained largely consistent across the dataset. Repetition time (TR) and echo time (TE) were relatively standardized. TR values were clustered into two groups (~700–800 ms and ~1700–2000 ms). TE values ranged from 2.8 to 3.8 ms. Field of view was uniformly 256 × 256 mm, with voxel sizes tightly grouped around $1.0 \times 1.0 \times 1.0$ mm³ or $0.5 \times 0.5 \times 1.0$ mm³.

Voxel resolution and image dimensions were well-conserved across the dataset. Most scans had identical image dimensions (240 × 240 × 180 mm³ or 240 × 240 × 90 mm³). Minor variations in slice count or voxel spacing did not meaningfully impact structural resolution. Since FD is derived from structural boundary complexity rather than absolute signal intensity, these small protocol differences are unlikely to introduce systematic bias into FD calculations. Additionally, all images underwent rigorous quality control, and subjects with significant artifacts or poor tissue contrast were excluded from analysis.

To quantify regional FD, whole-brain regions of interest (ROIs) were first mapped to patients' native brain space. Affine and deformable transformations were performed using FMRIB Software Library (FSL)[64,65] to register each patients' brain images into the Montreal Neurological Institute standard space[66]. The inverse warp of the deformable transformation was then applied to the 90-ROI parcellation map from the Automated Anatomical Labeling (AAL-90) atlas[67]. In Python, FD was calculated for each patients' ROIs warped to their native brain space. This approach of calculating FD over a defined region is necessary to capture self-similarity across spatial scales and ultimately meaningful complexity information. We used the box-counting method[68,69] which involves iteratively measuring the number of non-overlapping boxes, N, of incrementing sizes, s, needed to cover the surface of each ROI. The process starts by setting the largest box size to the length of the smallest dimension of the 3D image. The algorithm then systematically reduces the box size in a logarithmic fashion, halving the size at each step. At each box size, the number of boxes required to cover the non-zero voxels is counted. Finally, the relationship between the box size and the number of boxes needed is plotted on a logarithmic scale, and the slope of this plot, obtained through linear regression, provides the FD. As shown in Fig. 6B, FD was taken as the slope of the $\log(N)$ versus $\log(1/s)$ relationship.

### Validation of FD as a PD biomarker

Before evaluating the predictive value of regional FD, we sought to verify that the metric could discriminate individuals with PD from HCs and explain patient differences in disease severity. To reduce potential single-site bias and enhance study rigor, 70 HC datasets were sourced from three different studies and sites within OpenNeuro and open-source online collections[70–72]. Similarly, 70 PD datasets were selected from three different studies and sites[45,73], including 25 datasets randomly chosen from our primary UCSF cohort. To mitigate the effects of age on brain structure, HCs were age-matched to PD patients.

To ensure assumptions of normality were met for the residuals in regression analyses, we created quantile-quantile plots and performed the Anderson-Darling test, as detailed in the Supplementary Information (see Supplementary Fig. 1). Following normality testing, two-sided *t* tests were conducted to compare all 90 regional FD values across the PD and HC groups. False discovery rate correction for multiple

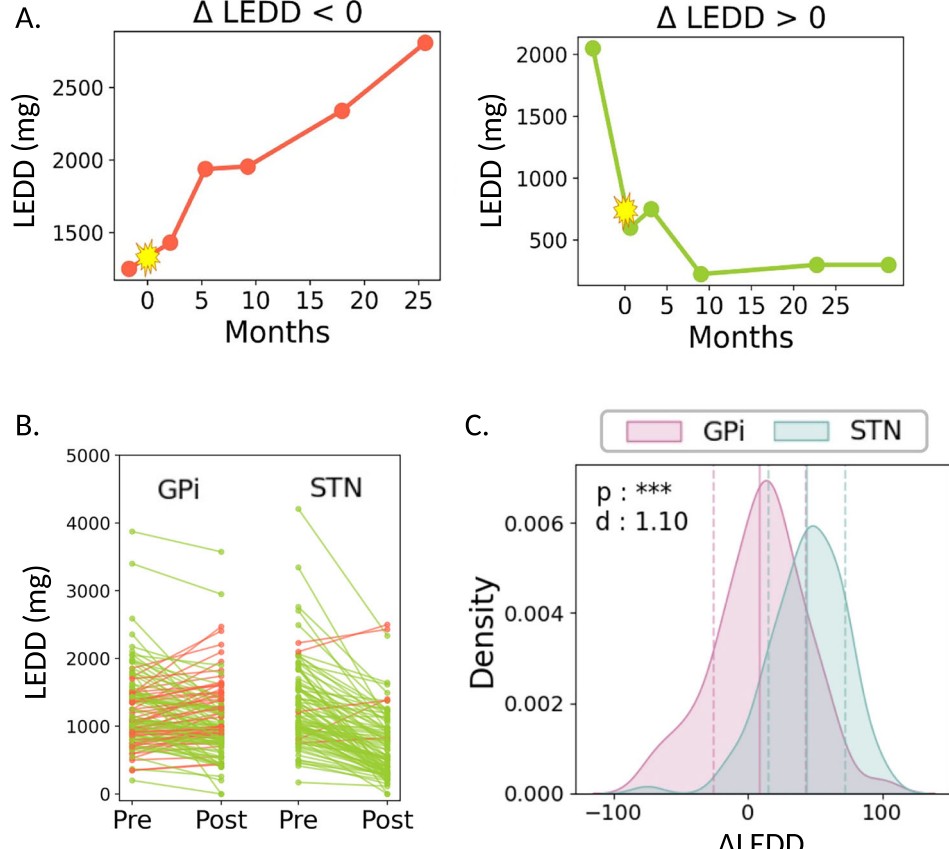

**Fig. 5 | Individual and target-specific changes in dopaminergic medications following DBS. A** Example LEDD trajectories over time for two patients: one with an increase in LEDD (ΔLEDD ≤ 0) and one with a decrease (ΔLEDD > 0). The zero-month mark in yellow indicates the date of DBS surgery. **B** To visualize general patterns, pre- and post-DBS LEDD values are shown for all patients (*n* = 231), split by DBS target (globus pallidus internus (GPi) vs. subthalamic nucleus (STN)). Each line represents one patient. Green lines indicate a decrease in LEDD following DBS; red lines indicate an increase. More red lines can be observed in the GPi group. **C** Plotted distributions of LEDD change were significantly different for STN (*n* = 96) and GPi (*n* = 135) patient groups (*t* test: *p* < 0.001, Cohen's *d* = 1.10), further demonstrating more negative values (less reduction in LEDD) for GPi-implanted patients. The mean and standard deviation of each distribution are denoted as a vertical solid line and two dotted lines, respectively. Source data are provided as a Source Data file.

comparisons was applied using the Benjamini–Hochberg method[74]. ROIs with an adjusted *p*-value < 0.05 were deemed statistically significant. The predictive power of FD in discerning between patients and HCs was determined by training and testing a classification HGNN model (see Data representation and hypergraph neural network classification subsection below).

Linear regression was thereafter used to evaluate the relationship between FD of 90 ROIs and disease severity in our primary cohort of 231 patients with PD. Raw total motor scores based on the preoperative MDS-UPDRS III exam were used as a proxy of disease severity, excluding rigidity and postural stability exam items (3.3 and 3.12) due to some patients undergoing virtual examination. The analysis was conducted using patients' total motor scores evaluated in the OFF- and ON-medication states, and age at MRI and sex were modeled as covariates. In total, 180 regression models were run to cover all the brain regions and medication states, and the Benjamini–Hochberg method was used to correct for multiple comparisons. Correlations with a *p*-value < 0.05 were deemed significant and 95% confidence intervals were calculated for regression coefficients to assess the precision of each estimate.

Finally, in addition to verifying that FD can discriminate PD from healthy individuals, we sought to demonstrate that FD is a reliable metric. Using a separate healthy control dataset from the Neuroimaging Tools and Research Collaboratory[75], we conducted a reproducibility analysis to ensure the stability of FD across repeated imaging sessions.

The detailed methods and results of this sub-analysis are outlined in the Supplementary Information (see Supplementary Fig. 2).

**Feature selection**

A multi-step statistical approach was employed to investigate whether additional variance in DBS outcomes could be explained by T1-w regional FD features. In addition to the 90 regional FD features, clinical features considered included: age at the time of MRI, sex, DBS target (STN or GPi), hemispheres treated (unilateral or bilateral), and pre-operative medication responsiveness quantified as the percent improvement in MDS-UPDRS III motor scores from the OFF- to ON-medication state.

Given the high dimensionality of the dataset, we first implemented feature selection to prevent model overfitting and enhance interpretability. Feature selection was performed using only the training dataset (~70%), to ensure that the validation (~15%) and testing (~15%) data remained independent for unbiased evaluation. To complement the main analysis, we conducted a cross-validation analysis to evaluate feature selection consistency across the entire dataset (see Supplementary Fig. 3). All continuous features were normalized to have a mean of zero and standard deviation of one such that features with more extensive range could not disproportionately influence the results[76].

Lease Absolute Shrinkage and Selection Operator (LASSO) regression was applied due to its superiority for feature selection

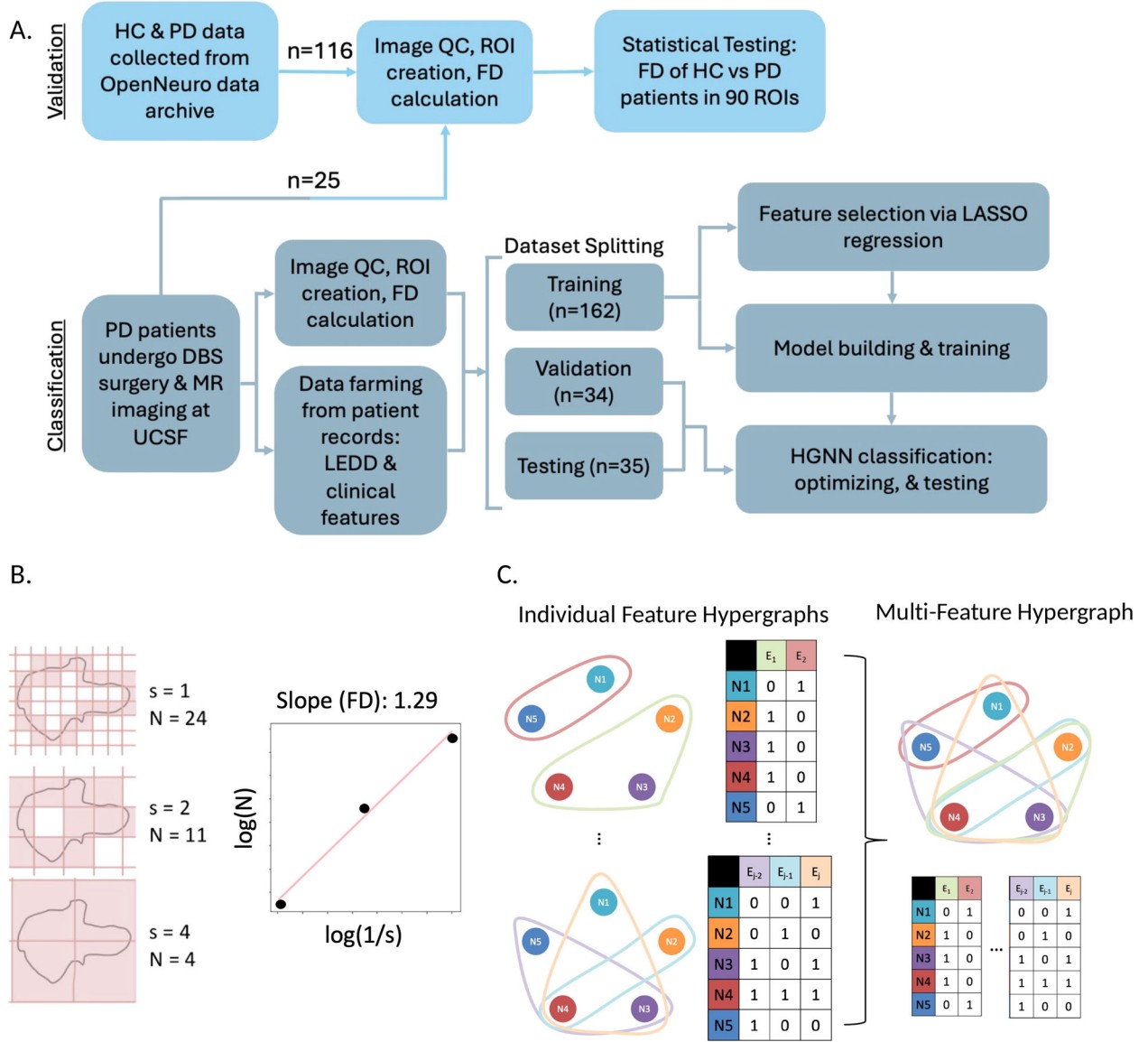

**Fig. 6 | Overview of study methods. A** Flow chart illustrating data sources and analysis steps for biomarker validation, variance analysis, and classification tasks. **B** Simple schematic of the box-counting method for calculating the fractal dimension of a surface. **C** Simple schematic of hypergraphs and their respective incidence matrices. Individual feature hypergraphs are concatenated horizontally to create a multi-feature hypergraph. $N_i$ = node/subject, $E_j$ = hyperedge.

within high-dimension datasets[77]. LASSO penalizes the absolute size of the regression coefficients, effectively shrinking some of them to zero, which simultaneously achieves feature selection and regularization. To select the optimal shrinkage factor ($\alpha$), we employed 5-fold cross-validation, selecting the $\alpha$ that maximized the ROC AUC of the classification model. Selected features provided a refined set of predictors for constructing our hypergraphs for variance analysis and classification learning tasks.

Ridge regression was thereafter performed to incorporate the complete set of LASSO-selected features while addressing multicollinearity and enhancing model stability[78]. Ridge regression, which penalizes the square of the regression coefficients, was chosen for its ability to handle multicollinearity by shrinking the coefficients of correlated predictors towards each other. This method yielded coefficients representing the relative importance of each selected feature in explaining variance in DBS outcomes. Normalized coefficients provided weights for our HGNN model described below as a pseudo

representation of feature importance in the combined multi-feature hypergraph structure.

## Variance analysis

We sought to quantify the additional variance in LEDD outcomes explained by FD features when combined with predictive clinical features. To achieve this, we performed a variance analysis comparing two OLS regression models. The first model included only clinical features as predictors, including: the DBS target (STN or GPi), hemisphere treated (unilateral or bilateral), age at the time of MRI, sex, and the preoperative change in MDS-UPDRS scores from the OFF- to ON-medication state (%Δ MDS-UPDRS III). The second model included all predictors from the first clinical model plus selected FD features derived from preoperative T1-w MRI scans.

Both models were fit using the OLS method, and respective R-squared values were compared to assess the proportion of variance explained by each model. To determine whether the inclusion of FD

significantly improved the model's explanatory power, we performed an F-test for nested models. The F-test compared the fit of the two models by testing the null hypothesis that adding imaging features would not improve the model significantly over the clinical features alone. Specifically, the F-statistic was calculated as

$$F = \frac{\frac{RSS_1 - RSS_2}{df_2 - df_1}}{\frac{RSS_2}{df_2}} \qquad (2)$$

where $RSS_i$ and $df_i$ are the individual models' residual sum of squares and degrees of freedom, respectively. The resulting F-statistic was used to calculate the *p*-value indicating whether additional variance explained by the imaging features was statistically significant.

### Data representation and hypergraph neural network classification

We constructed hypergraph structures based on the LASSO-selected features to capture the complex relationships between features. Each feature formed a hypergraph, with nodes representing subjects and hyperedges representing similarity or categorical grouping within the feature space.

For categorical features such as DBS target (STN vs. GPi), hemispheres implanted (unilateral vs. bilateral), and sex, hyperedges were formed by grouping subjects based on their category. For continuous variables such as preoperative motor scores and regional FD, we created k-uniform hypergraphs with k varying from 5 to 13 to assess the impact of different connectivity scales. This range of k-values was chosen empirically to ensure stable results to the point of plateau. The k-uniform hypergraphs were constructed by connecting each node to its k-1 nearest neighbors based on feature similarity. The edges of all hypergraphs were weighted using the normalized coefficients from ridge regression that reflect relative feature importance. The hypergraphs of individual features were concatenated horizontally, as shown in Fig. 6C, to create multi-feature hypergraphs. Two hypergraphs were constructed, one with only clinical features and one that contained clinical features and FD features. Both hypergraphs were used to independently train the classification model to test if the addition of FD improved predictive performance and by how much.

The construction of hypergraphs was implemented using the DeepHyperGraph PyTorch package[79,80], which provides efficient tools for HGNN construction and training. HGNNs are designed to process data structured as hypergraphs. In the HGNN framework, the key components include hyperedge convolution layers, node-edge-node transformations, and spectral convolution operations. Spectral convolution is at the core of the HGNN architecture, where signals defined on hypergraphs are convolved with learnable filters to extract features that capture high-order correlations inherent in high-dimensional data.

Our HGNN model was comprised of multiple layers, each responsible for transforming node features through convolutional operations. Initially, node features were processed by learnable filter matrices, resulting in feature representations that captured local and global structural information within the hypergraph. Hyperedge features were then derived through aggregation mechanisms that leveraged the hyperedge connectivity information encoded in the multi-feature hypergraph incidence matrix. This node-edge-node transformation process allowed our HGNN model to effectively capture and propagate information across different scales of connectivity within the hypergraph, facilitating the extraction of discriminative features for downstream classification tasks[81]. The final convolutional layer performed classification to predict the likelihood of each subject's ΔLEDD being less than or equal to 0, or greater than 0. The model was trained using a supervised learning approach, with cross-entropy loss as the objective function.

Model hyperparameters were optimized using the validation dataset, and performance was monitored at each epoch using the validation subset to identify the best-performing model. The final model performance was evaluated on the test subset. The ROC AUC was calculated for the clinical features hypergraph model and the combined model with both clinical and FD features to assess differences in classification performance and overall model performance.

### Target-specific differences in FD predictions

We conducted an interaction analysis to investigate whether the relationship between brain structure, as measured by FD, and post-DBS medication change differed by DBS target (STN vs. GPi). This analysis aimed to identify target-specific effects on outcomes by including interaction terms in our regression models. Specifically, interaction terms were created between FD values for each of the 16 LASSO-selected ROIs and the binary variable indicating the DBS target (STN or GPi). For each ROI, we fitted an OLS regression model where the dependent variable was the pre-to-post-DBS ΔLEDD. Predictor variables included the FD of the region, covariates (age, sex, hemispheres treated, percent improvement in preoperative motor scores), and an interaction term between FD and the DBS target. The interaction term allowed us to assess whether the effects of FD on DBS outcomes vary depending on the stimulation target.

The statistical significance of the interaction terms was assessed using *p*-values ($p < 0.05$) to determine whether the association between FD and ΔLEDD was significantly different between the two target groups. Additionally, we plotted interaction effects to visually represent the relationships between FD, DBS target, and ΔLEDD. Interaction plots illustrate how the impact of FD on outcomes changes for STN and GPi patients, thus enabling direct comparison of these target-specific effects.

### Reporting summary

Further information on research design is available in the Nature Portfolio Reporting Summary linked to this article.

## Data availability

In line with promoting transparency and reproducibility in scientific research, tabular source data used for all figures and analyses in this study, including regional fractal dimension values and clinical features, are publicly available on GitHub: https://github.com/Radiology-Morrison-lab-UCSF/T1w_FractalDimension and on Code Ocean (DOI:10.24433/CO.9780346.v1). A curated subset of the raw MRI data will be made publicly available via OpenNeuro pending IRB modification and updated participant consent. Due to the retrospective nature of the dataset and the lack of open-sharing provisions in earlier consent forms, some imaging data cannot be made publicly available. Tabular processed data and analysis templates provide full transparency and reproducibility of the results presented. Source data are provided with this paper.

## Code availability

The above GitHub and Code Ocean repositories also includes a de-identified example T1-weighted MRI scan and corresponding FD values for 90 ROIs from the AAL atlas, enabling users to reproduce at least one FD calculation for a single subject. Also included is code for image processing workflows, fractal dimension calculation, hypergraph classification for neuroimaging datasets, and statistical analysis pipelines.

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

## Acknowledgements

This work was supported by the National Institute of Neurological Disorders and Stroke (R01NS130066, awarded to M.A.M.). The authors thank the patients and their families, as well as the clinical staff at UCSF for their support.

## Author contributions

D.S. contributed to study conception, data collection, data analysis, data interpretation, and manuscript writing and revision; S.D., J.M. and S.W. contributed to data collection and manuscript revision; J.K. contributed to data analysis, data interpretation, and manuscript revision; P.A.S. and D.D.W. contributed to resources, data interpretation, and manuscript revision; J.L.O. and I.O.B. contributed to study conception, resources, data interpretation, and manuscript revision; M.A.M. contributed to study conception, resources, data collection and interpretation, manuscript writing and revision, and supervision.

## Competing interests

P.A.S. receives support from Medtronic and Boston Scientific for fellowship education. J.L.O. receives support from Medtronic and Boston Scientific for research and education and consults for AbbVie. D.D.W. receives support from Boston Scientific, and consults for Boston Scientific, Medtronic, and Iota Biosciences. D.S., S.D., J.M., S.W., J.K., I.O.B. and M.A.M. declare no competing interests.

## Additional information

**Supplementary information** The online version contains
supplementary material available at

Melanie A. Morrison.

**Peer review information** *Nature Communications* thanks Muthuraman
Muthuraman and the other, anonymous, reviewer(s) for their contribu-
tion to the peer review of this work. A peer review file is available.

**Devin Schoen** [1,2], **Skyler Deutsch** [1], **Juhi Mehta** [1], **Sarah Wang**[3], **John Kornak**[4], **Philip A. Starr** [2,5], **Doris D. Wang** [2,5],
**Jill L. Ostrem**[3], **Ian O. Bledsoe**[3] & **Melanie A. Morrison** [1,2] ✉

[1]Department of Radiology and Biomedical Imaging, University of California, San Francisco, USA. [2]UCSF-UC Berkeley Joint PhD Program in Bioengineering,
Berkeley, USA. [3]Department of Neurology, University of California, San Francisco, USA. [4]Department of Epidemiology and Biostatistics, University of Cali-
fornia, San Francisco, USA. [5]Department of Neurological Surgery, University of California, San Francisco, USA. ✉e-mail: melanie.morrison@ucsf.edu

