## [Transparent Peer Review file · Nature Communications]

Boundary complexity of cortical and subcortical areas predicts deep brain stimulation outcomes in Parkinson's disease

Corresponding Author: Dr Melanie Morrison

Version 0:

Reviewer comments:

Reviewer #1

(Remarks to the Author)

The manuscript is well-written and organized. However, I have concerns regarding the premise and methodology. The treatment of Parkinson's disease (PD) with deep brain stimulation (DBS) operates primarily at a functional level. Predicting outcomes based solely on static structural imaging lacks comprehensive evaluation of the disease's functional status. I believe that functional disorders are better illustrated through functional measurements, such as fMRI, PET, or EEG, with structural imaging serving as an auxiliary modality.

Introduction

Could the authors elaborate on the biological mechanisms of FD (feature deformation)? Considering features like cortical thickness or surface changes are associated with disease progression and treatment response, how do the authors emphasize the unique contributions of FD in this context?

Methods

I believe that ROI-level computations may be less informative compared to voxel-level analyses. The division of brain regions appears arbitrary and might not capture detailed variations. What are the distinctions between FD and Jacobian determinants? Both represent deviations from normal brain anatomy—how do the authors differentiate their utility? Surgical outcomes in DBS are significantly influenced by variables such as electrode placement, stimulation contact points, and parameter settings. How did the authors account for or exclude these covariates, which might affect the response?

Results

The authors used LASSO regression for selecting neuroimaging and clinical features and OLS for selecting FD features. Why were two different models employed, and do they produce equivalent predictive outcomes?

An R^2 value of 0.388 is relatively low and indicates limited predictive performance.

An accuracy of 0.76 for a binary classification model is unconvincing and may not meet the threshold for robust clinical prediction.

Discussion

The interpretations of FD values in both subcortical and cortical areas appear speculative and require stronger evidence to support their claims.

(Remarks on code availability)

NA

Reviewer #2

(Remarks to the Author)

This paper represents a significant step forward in understanding the structural correlates of DBS outcomes in PD. The

methodological rigor and innovative use of FD as a biomarker are commendable. However, further research is needed to address the limitations of retrospective design, refine the predictive models, and validate findings in diverse clinical settings. I have pointed out the major and minor points which need to be revised for each section separately:

1. Introduction

The paper provides a clear and well-articulated introduction, highlighting the variability in Deep Brain Stimulation (DBS) outcomes in Parkinson's Disease (PD) and the need for improved prognostic criteria. The use of fractal dimension (FD) as an innovative biomarker is justified, with references to prior studies on brain complexity and neurodegeneration.

- However, the introduction could benefit from a more detailed explanation of why FD, specifically, may outperform other imaging biomarkers in predicting DBS outcomes.

2. Methodology

The study employs a robust methodological approach, integrating large-scale, single-center imaging data (231 patients) with advanced statistical and machine learning techniques. Key strengths include:

- The use of T1-weighted MRI, a widely available imaging modality, making the findings clinically translatable.
- Validation of FD as a biomarker through external datasets, enhancing the generalizability of results.
- The application of hypergraph neural networks (HGNNs), which are innovative and capable of capturing complex relationships among features.

However, the methodology has some limitations:

- The use of retrospective data and the reliance on changes in medication burden (Δ LEDD) as a proxy for DBS efficacy may not capture the full spectrum of patient outcomes, including motor and non-motor symptoms.
- The variability in MRI acquisition parameters over 13 years introduces potential noise, which could affect the reliability of FD calculations despite the efforts to control for this.

3. Results

The results are presented comprehensively, with appropriate statistical tests and visualizations. Key findings include:

- FD differences between PD patients and healthy controls (HCs) in specific brain regions.
- The additional predictive value of FD features, explaining 13.6% more variance in DBS outcomes compared to clinical features alone.
- Superior classification performance of the combined model (clinical + FD features) over clinical features alone.

While promising, some aspects could be improved:

- The reported classification accuracy (AUC = 0.76) suggests moderate predictive power, leaving room for improvement, possibly through multimodal imaging or inclusion of additional biomarkers.
- The subgroup analysis for STN vs. GPi DBS targets is relegated to supplementary material. Greater emphasis on these target-specific effects would enhance clinical relevance.

4. Discussion

The discussion effectively situates the findings within the broader literature, addressing the biological plausibility of FD as a biomarker for PD progression and DBS outcomes. The exploration of cortical vs. subcortical FD changes is insightful, but:

- Contradictions with previous studies (e.g., Li et al.) are acknowledged but not fully resolved, warranting further exploration.
- The potential role of non-motor regions (e.g., amygdala, temporal pole) in influencing DBS outcomes is an interesting avenue for future research but could be expanded upon in this paper.

However, additional limitations could be discussed, such as:

- The relatively low specificity of FD changes to PD, which may limit its utility in differentiating DBS candidates from other patient populations.
- Potential confounding effects of demographic and clinical factors not fully captured in the model.
- Prospective validation with standardized imaging protocols and comprehensive outcome measures.
- Exploration of the cost-effectiveness and scalability of incorporating FD analysis into routine clinical workflows.

(Remarks on code availability)

More commenting on the codes would be helpful for the reviewer to judge it.

Reviewer #3

(Remarks to the Author)

This revised manuscript explores the prediction of medication changes in Parkinson's disease (PD) patients after deep brain stimulation (DBS) treatment based on the fractal dimension (FD) in T1-weighted magnetic resonance imaging (MRI) data. The revised manuscript used FD as a predictor of DBS treatment outcome in PD patients for the first time and validated the correlation between FD and PD disease severity. The revised manuscript used publicly available datasets from multiple centres for the validation of FD, enhancing the generalisability and reliability of the findings. The revised manuscript used both traditional clinical features (e.g., age, gender, DBS electrode location, etc.), and imaging features, such as FD, to construct a multi-feature prediction model, and performed the classification learning task via a hypergraphical neural network (HGNN), which better captures the complex relationships between features to improve the performance of the prediction model.

The revised version also has some problems :

1. the description of subgraphs in the figure notes of Fig. 1 is not clear enough, perhaps it could be consistent with the figure notes of other figures.
2. the text of the tables in the revised version overlaps and is not legible.
3. the data used in the revised draft is from a different scanner and protocol, there may be data quality and consistency issues, how can this be resolved? Although the article argues that such data brings diversity and enhances the

generalisation of the model, it may also introduce noise, how did the authors distinguish and address this?
4. Although the authors used multicentre data, the main analysis in the revised manuscript was based on 231 patients from a single centre. The sample size and diversity of the data may not be sufficiently representative of the broad population of PD patients, especially those with different disease duration and symptom severity.

(Remarks on code availability)

Version 1:

Reviewer comments:

Reviewer #1

(Remarks to the Author)

Thank you for authors' detailed and thoughtful responses. I appreciate the effort your team has put into addressing my concerns, and I find that most of my initial concerns have been adequately resolved. And sorry for my Comment 7, OLS is not for features selection. I should not mention it.

(Remarks on code availability)

Reviewer #2

(Remarks to the Author)

The authors have adequately answered my queries in the revised version. I do not have further comments on the revised manuscript.

(Remarks on code availability)

The code could contain more documentation (commented).

Reviewer #3

(Remarks to the Author)

I have no further comments.

(Remarks on code availability)

Null

made.

Manuscript ID: NCOMMS-24-78532

Title: BOUNDARY COMPLEXITY OF (SUB-) CORTICAL AREAS PREDICT DEEP BRAIN STIMULATION OUTCOMES IN PARKINSON'S DISEASE

Dear Reviewers,

We appreciate the time and effort that you have taken to evaluate our manuscript. We are grateful for the insightful comments, which have helped us strengthen our work. Below, we provide a point-by-point response to each comment, detailing the revisions made. **The reviewers' comments are reproduced in black, our responses are in blue, and changes in the manuscript are in red.**

We have also uploaded a revised manuscript where all changes are shown in **red font**, as requested.

Reviewer #1 (Remarks to the Author):

Comment 1:

The manuscript is well-written and organized. However, I have concerns regarding the premise and methodology. The treatment of Parkinson's disease (PD) with deep brain stimulation (DBS) operates primarily at a functional level. Predicting outcomes based solely on static structural imaging lacks comprehensive evaluation of the disease's functional status. I believe that functional disorders are better illustrated through functional measurements, such as fMRI, PET, or EEG, with structural imaging serving as an auxiliary modality.

Response:

We appreciate the reviewer's concern regarding the use of structural imaging for outcome prediction in a functional disorder. However, the fractal dimension (FD) metric reflects structural brain integrity, which serves as a surrogate for disease severity, thereby capturing an aspect of PD pathology that influences DBS outcomes. Our underlying hypothesis is that structural brain health influences functional response to DBS, as structure and function are intrinsically linked. Further supporting our approach is a recent 2023 review paper (*Andrews et al., 2023*) highlighting prior studies that have used structural MRI markers to explain DBS motor outcomes, including morphometric markers like FD.

Changes in manuscript:

We have provided additional rationale for using FD including our hypothesis and citations for two new supporting papers.

- Introduction: Lines 46-56

Introduction

Comment 2:

Could the authors elaborate on the biological mechanisms of FD (feature deformation)?

Response:

In the context of PD, FD reflects both cortical remodeling and subcortical degeneration, which are hallmarks of disease progression. Below we provide more detail on each of these characteristics captured by FD.

- Cortical remodeling & compensatory changes: FD can capture increases in cortical complexity that may arise due to compensatory reorganization. As dopaminergic neurodegeneration progresses, the cortex may exhibit increased gyrification and complexity, reflecting structural adaptations attempting to maintain functional integrity.
- Subcortical structural degeneration: In contrast, FD in subcortical regions such as the basal ganglia, putamen, and pallidum tends to decrease, mirroring known neurodegenerative processes in PD. The loss of dopaminergic neurons in the substantia nigra leads to structural simplification in these regions, resulting in reduced FD.
- Self-similarity in biological structures: FD is uniquely sensitive to the scale-invariant nature of brain morphology, capturing patterns of structural integrity across different spatial scales. This makes it complementary to traditional morphometric measures like cortical thickness, which may not fully encapsulate complex changes in neurodegeneration.

Changes in manuscript:

We have described the biological mechanisms of FD in the introduction. Our discussion also elaborates on this.

- Introduction: Lines 46-52

Comment 3:

Considering features like cortical thickness or surface changes are associated with disease progression and treatment response, how do the authors emphasize the unique contributions of FD in this context?

Response:

While cortical thickness and volumetric analyses have been widely used in Parkinson's disease (PD), FD may be more sensitive to subtle structural abnormalities, capturing complexity changes that are not well reflected in traditional measures. For example, cortical thickness provides a one-dimensional measure of atrophy, whereas FD quantifies complexity across multiple spatial scales, making it more sensitive to structural disruptions that do not necessarily manifest as volumetric loss. FD has been used to assess cortical folding patterns and structural integrity, providing complementary information to traditional atrophy measures (*Im et al., 2006; Kiselev et al., 2003*). As Smith et al. (2019) discussed, FD provides an important tool for assessing morphological changes beyond conventional volumetric measures, making it a valuable metric for studying neurodegenerative disorders. Studies in Alzheimer's disease have even suggested that FD alterations can precede detectable cortical thinning (*King et al., 2010*).

Changes in manuscript:

We have emphasized the unique contributions of FD compared to conventional thickness and volume measures, including citations for three new supporting papers.

- Introduction: Lines 46-52

Methods

Comment 4:

I believe that ROI-level computations may be less informative compared to voxel-level analyses. The division of brain regions appears arbitrary and might not capture detailed variations.

Response:

While voxel-based approaches offer fine-grained detail, FD must be measured within a region of interest (ROI) to appropriately capture fractal properties. FD was originally developed in the context of natural structures, such as coastlines and geological formations, and has since been adopted for medical imaging to quantify structural complexity in biological systems. Just as the complexity of natural formations is measured over a defined area rather than at single points, FD calculations in the brain rely on regional computations rather than individual voxel-based estimates. While ROI selection inherently involves some level of anatomical parcellation, we have chosen a widely used atlas to minimize arbitrariness and improve reproducibility across studies.

Changes in manuscript:

We have emphasized the need to calculate FD over a defined region of interest.

- **Methods: Lines 372-374**

Comment 5:

What are the distinctions between FD and Jacobian determinants? Both represent deviations from normal brain anatomy—how do the authors differentiate their utility?

Response:

While both metrics assess deviations from normal brain anatomy, they capture distinct aspects of structural change and are used for different analytical purposes:

- FD quantifies the complexity and self-similarity of brain structures, providing a scale-invariant measure of morphological irregularities. It captures how intricate a structure is across multiple spatial resolutions, making it well-suited for assessing neurodegenerative processes that involve changes in cortical folding, gyrification, and subcortical morphology. FD is particularly useful for characterizing changes in brain organization that may not correspond directly to local volumetric expansion or contraction.
- Jacobian Determinants measure local volumetric changes that occur during non-linear image registration. They quantify regional expansions and contractions, typically used to assess atrophy or tissue displacement. Unlike FD, which describes shape complexity, Jacobian determinants reflect localized structural deformations relative to a template.

Thus, while both metrics describe structural differences, FD provides information about morphological complexity and organization, whereas Jacobian determinants track absolute volumetric changes. These are complementary rather than interchangeable measures.

Changes in manuscript:

As this distinction is methodologically clear, we have not made any modifications to the manuscript.

Comment 6:

Surgical outcomes in DBS are significantly influenced by variables such as electrode placement, stimulation contact points, and parameter settings. How did the authors account for or exclude these covariates, which might affect the response?

Response:

The goal of this study is to aid pre-surgical prediction by evaluating the influence of brain health on DBS outcomes before any information on electrode placement, stimulation parameters, or contact selection is available. This is a key distinction from studies that focus on post-operative optimization—our primary objective is to determine whether pre-surgical structural brain features can help predict DBS response and identify good candidates for the procedure. We acknowledge that post-operative factors such as lead location and stimulation parameters significantly impact DBS outcomes. A separate study is currently underway to analyze post-operative lead placement and stimulation settings, aiming to disentangle the relative contributions of structural brain health versus surgical factors in DBS efficacy.

Changes in manuscript:

We have restated our goal of true pre-operative prediction and highlighted separate ongoing work evaluating the relative impact of brain health versus surgical factors on DBS outcomes.

- Discussion: Lines 293-297

Results**Comment 7:**

The authors used LASSO regression for selecting neuroimaging and clinical features and OLS for selecting FD features. Why were two different models employed, and do they produce equivalent predictive outcomes?

Response:

We would like to clarify that LASSO regression was used for feature selection across all neuroimaging and clinical features, including FD. OLS regression was not used for feature selection but rather for final model evaluation and statistical inference once the most predictive features were identified. This two-step approach is standard practice in high-dimensional regression analyses:

- LASSO Regression was applied to select the most relevant features by introducing sparsity, which helps avoid overfitting when working with multiple predictors.
- OLS Regression was then used only on the selected features to assess their individual contributions and test predictive relationships within a more interpretable framework.

Additionally, we confirm that all statistical methods were reviewed by our statistician Dr. John Kornak, PhD, who is a co-author on this manuscript.

Changes in manuscript:

No changes have been made as the manuscript in its current form states that LASSO regression was used for feature selection and OLS regression for model evaluation.

Comment 8:

An R^2 value of 0.388 is relatively low and indicates limited predictive performance. An accuracy of 0.76 for a binary classification model is unconvincing and may not meet the threshold for robust clinical prediction.

Response:

This study represents one piece of the larger effort to improve DBS outcome prediction. This model is designed as a first step, focusing on pre-surgical structural brain health and laying the foundation for future multimodal models. While an R^2 of 0.388 and an accuracy of 0.76 indicate moderate predictive power, it is important to contextualize these values:

- This model improves upon current clinical baselines, including MDS-UPDRS-based predictions, demonstrating that FD features add meaningful information beyond standard clinical measures.
- This is a conservative, single-center model, and we anticipate further improvements as more T1-weighted MRI data are collected, multicenter data are incorporated, and additional modalities (e.g., diffusion MRI, functional MRI) are integrated.
- Clinical decision-making rarely relies on a single predictive factor, and this model is not intended as a standalone clinical tool but rather as part of a comprehensive, multimodal predictive framework that will continue to be refined.

Changes in manuscript:

We have reiterated the clinical significance of our model and anticipated improvements with multicenter and multimodal data integration.

- Discussion: Lines 256-260

Discussion

Comment 9:

The interpretations of FD values in both subcortical and cortical areas appear speculative and require stronger evidence to support their claims.

Response:

Our findings—increased FD in cortical regions and decreased FD in subcortical structures in PD patients compared to healthy controls—align with emerging literature on morphological complexity changes in neurodegeneration. While the subcortical atrophy process in PD is well known, we understand that our cortical findings of increased FD, could be viewed as speculative given limited studies. This is just a hypothesis as we state in our discussion. However, encouragingly, after surveying the literature further we were able to find three additional studies (*Bullmore et al., 1994, Wolf et al., 2021, Li et al., 2024*), including two PD cohorts, that also reported instances of regionally increased in FD. The very recent *Li et al. 2024* paper demonstrating increased temporal lobe FD in patients with PD and hyposmia was a particularly fascinating find to us, as we observed a double-peak distribution of FD in PD olfactory areas (Fig 2B) and had hypothesized that it could be a biomarker of hyposmia. By disseminating this work, we hope to promote discourse around these interpretations which could lead to prospective studies that help us better understand cortical complexity in disease.

Changes in manuscript:

We have included citations for three new supporting papers.

Discussion: Lines 144-161

Reviewer #1 (Remarks on code availability):

NA

Reviewer #2 (Remarks to the Author):

This paper represents a significant step forward in understanding the structural correlates of DBS outcomes in PD. The methodological rigor and innovative use of FD as a biomarker are commendable. However, further research is needed to address the limitations of retrospective design, refine the predictive models, and validate findings in diverse clinical settings. I have pointed out the major and minor points which need to be revised for each section separately:

1. Introduction

The paper provides a clear and well-articulated introduction, highlighting the variability in Deep Brain Stimulation (DBS) outcomes in Parkinson's Disease (PD) and the need for improved prognostic criteria. The use of fractal dimension (FD) as an innovative biomarker is justified, with references to prior studies on brain complexity and neurodegeneration.

Comment 1:

However, the introduction could benefit from a more detailed explanation of why FD, specifically, may outperform other imaging biomarkers in predicting DBS outcomes.

Response:

The goal of this study is not to position FD as a replacement or superior alternative to other imaging biomarkers (e.g., functional MRI, diffusion MRI), but rather to demonstrate its value as a complementary structural feature that can be integrated into broader predictive models alongside other clinical and imaging predictors. However, compared to other conventional morphometric features (e.g., cortical thickness or volumetric atrophy), FD is in fact superior in that it uniquely captures structural complexity; we have discussed this in our responses to Reviewer 1, comments 2 & 3.

Changes in manuscript:

We have emphasized that FD is viewed as a complementary feature. We have also emphasized the unique contributions of FD compared to conventional thickness and volume measures.

- Introduction: Lines 43-44
- Introduction: Lines 46-52

2. Methodology

The study employs a robust methodological approach, integrating large-scale, single-center imaging data (231 patients) with advanced statistical and machine learning techniques. Key strengths include:

- The use of T1-weighted MRI, a widely available imaging modality, making the findings clinically translatable.
- Validation of FD as a biomarker through external datasets, enhancing the generalizability of results.
- The application of hypergraph neural networks (HGNNs), which are innovative and capable of capturing complex relationships among features.

Comment 2:

However, the methodology has some limitations: The use of retrospective data and the reliance on changes in medication burden (Δ LEDD) as a proxy for DBS efficacy may not capture the full spectrum of patient outcomes, including motor and non-motor symptoms.

Response:

We acknowledge this limitation; however, these were the only data available to us at this time. Δ LEDD is a widely used proxy for DBS efficacy: while it does not capture the full range of motor and non-motor outcomes, it remains a clinically relevant and objective measure of post-DBS treatment response and broadens the applicability of our models (since most centers have access to T1 and Δ LEDD data). That said, we recognize the importance of capturing a broader spectrum of patient outcomes, including motor (MDS-UPDRS III) and non-motor (quality of life, neuropsychological measures) response. To address this, we have NIH R01 funding to acquire prospective data that includes these standardized clinical outcomes. This next phase of research will allow us to refine and enhance predictive modeling approaches by incorporating more comprehensive DBS outcome metrics beyond Δ LEDD.

Changes in manuscript:

We have emphasized the limitations and strengths of our current approach. We have also highlighted ongoing prospective collection of standardized clinical outcomes to improve our model predictions and their clinical impact.

- Discussion: Lines 278-285

Comment 3:

The variability in MRI acquisition parameters over 13 years introduces potential noise, which could affect the reliability of FD calculations despite the efforts to control for this.

Response:

While differences in imaging parameters over time are an important factor to consider, we believe that this variability also enhances the generalizability of our model. Specifically, it demonstrates that FD can be extracted from clinically heterogeneous data—a necessary step toward real-world application. Despite the retrospective nature of the dataset, our imaging acquisition parameters were remarkably consistent, with the majority of scans falling into two primary clusters:

- TR values clustered into two primary groups (~700–800 ms and ~1700–2000 ms) with a large proportion of subjects having TR within $\pm 5\%$ of the two most common values
- TE was highly uniform, ranging from 2.8–3.8 ms
- Voxel size and image dimensions were also stable and highly clustered, with most scans falling into 240x240 (1x1x1 mm³) or 512x512 (0.5x0.5x1 mm³)

While minor variations in TR (680–2000 ms) exist, these differences are unlikely to introduce substantial bias in FD measurements. The T1-weighted sequences used in this study primarily rely on contrast differences driven by tissue relaxation properties. Given the relatively short TR range in this cohort, all scans remain within a conventional T1-weighted contrast regime. FD calculations are dependent on image resolution and tissue boundary clarity. Since most scans have similar voxel sizes and contrast, we would expect our FD measurements to remain robust across different scanner protocols. To further demonstrate the reliability of our FD calculation, a new reproducibility analysis has been conducted using a publicly-available longitudinal imaging data, the results of which are now included in our supplementary materials (please see our response to Reviewer 3, comment #3).

Changes in manuscript:

We have revised the methods section to clarify this point and reassure readers that the majority of patients were scanned under consistent parameters.

- Methods: Lines 352-365

3. Results

The results are presented comprehensively, with appropriate statistical tests and visualizations.

Key findings include:

- FD differences between PD patients and healthy controls (HCs) in specific brain regions.
- The additional predictive value of FD features, explaining 13.6% more variance in DBS outcomes compared to clinical features alone.
- Superior classification performance of the combined model (clinical + FD features) over clinical features alone.

While promising, some aspects could be improved:

Comment 4:

The reported classification accuracy (AUC = 0.76) suggests moderate predictive power, leaving room for improvement, possibly through multimodal imaging or inclusion of additional biomarkers.

Response:

We agree with this reviewer comment and have already addressed this in our response to Reviewer 1, comment #8.

Changes in manuscript:

We have reiterated the clinical significance of our model and anticipated improvements with multicenter and multimodal data integration.

- Discussion: Lines 256-260

Comment 5:

The subgroup analysis for STN vs. GPi DBS targets is relegated to supplementary material. Greater emphasis on these target-specific effects would enhance clinical relevance.

Response:

Given the clinical relevance of this distinction, we have moved the subgroup analysis from the supplementary material to the main manuscript and expanded the discussion to better contextualize the findings.

Changes in manuscript:

We have moved these results from the supplementary materials to the main manuscript, including the figure which is now Figure 3. Other line changes include:

- Results: Lines 107-119
- Methods: Lines 516-533
- Discussion: Lines 207-215

4. Discussion

The discussion effectively situates the findings within the broader literature, addressing the biological plausibility of FD as a biomarker for PD progression and DBS outcomes. The exploration of cortical vs. subcortical FD changes is insightful, but:

Comment 6:

Contradictions with previous studies (e.g., Li et al.) are acknowledged but not fully resolved, warranting further exploration.

Response:

While our findings (when comparing FD for PD vs. controls) align with some prior research (*Kubera et al.*) and diverge from others (*Li et al.*), fully resolving this discrepancy is beyond the scope of this manuscript. Rather, we aim to contribute additional evidence to the ongoing discourse surrounding FD as a neuroimaging marker and as a new predictor of DBS outcomes. Our validation sub-analyses further support the robustness of our approach, though we do recognize that continued investigation is necessary to fully reconcile differences across studies.

Changes in manuscript:

We have clarified this point in the discussion to ensure that our intent is clear.

- Discussion: Lines 163-173

Comment 7:

The potential role of non-motor regions (e.g., amygdala, temporal pole) in influencing DBS outcomes is an interesting avenue for future research but could be expanded upon in this paper.

Response:

While the classical view of DBS efficacy primarily focuses on motor circuits, studies have suggested that the involvement of non-motor may play a role in individual variability in response to DBS. Cognitive and psychiatric symptoms are increasingly recognized as important factors influencing post-DBS outcomes, with studies suggesting that PD patients with pre-existing cognitive impairment or neuropsychiatric symptoms may experience less favorable results following DBS. To clarify this point, we have expanded our discussion to incorporate prior research on the interaction between non-motor network involvement and DBS outcomes.

Changes in manuscript:

We have noted in our discussion the association between cognitive and psychiatric symptoms and DBS outcomes, including two new citations from supporting papers.

- Discussion: Lines 233-235

Comment 8:

However, additional limitations could be discussed, such as: The relatively low specificity of FD changes to PD, which may limit its utility in differentiating DBS candidates from other patient populations.

Response:

We acknowledge this shortcoming regarding the specificity of FD changes to PD; however, it was not our intention to use FD as a diagnostic marker to differentiate PD from other neurological conditions. We performed the PD vs. HC classification sub-analysis to confirm that FD exhibits meaningful differences in PD generally, supporting its biological relevance as a disease marker.

Changes in manuscript:

Given that the specificity of FD for disease classification lies outside the scope of our study and the rationale for the PD vs. HC sub-analysis is already explained in the manuscript, we have not made any further changes.

Comment 9:

Potential confounding effects of demographic and clinical factors not fully captured in the model.

Response:

We sought to minimize these influences by incorporating key variables (age, sex, baseline MDS-UPDRS scores, DBS target, hemispheres treated) as covariates. Other factors (such as detailed medication history, psychiatric comorbidities, or subtle variations in disease phenotype) were not readily extractable from our decade-long retrospective dataset. Fortunately, a new study that emerged following our initial submission (*Dehghan et al., 2024*) showed that our imaging metric FD may be sensitive to structural differences in PD phenotypes that we were unable to account for using clinical data. Indeed, it is exactly our goal to select for MRI features that model the characteristics of a disease subtypes or stage in an objective manner to more accurately predict DBS outcomes. As our model complexity increases through multimodal data integration (e.g. adding fMRI data), we expect to improve on this goal by way of increasing our prediction accuracy.

Changes in manuscript:

We have highlighted this as a limitation and included a citation for the new supporting paper.

- Discussion: Lines 291-293

Comment 10:

Prospective validation with standardized imaging protocols and comprehensive outcome measures.

Response: We agree with the reviewer that prospective validation with standardized imaging protocols and comprehensive outcome measures is an important next step and is the focus of our ongoing R01 research. This point has been discussed in multiple sections of the manuscript.

Changes in manuscript:

We have emphasized this need throughout the manuscript.

- Discussion Lines 256-260; 282-285; 297-303

Comment 11:

Exploration of the cost-effectiveness and scalability of incorporating FD analysis into routine clinical workflows.

Response:

FD offers several advantages that make it feasible for routine clinical use:

- FD can be computed from standard T1-weighted MRI.
- The computational processing required for FD extraction is lightweight and can be performed on standard clinical workstations, making it accessible without the need for high-performance computing infrastructure.
- FD extraction can be fully automated within existing image-processing pipelines, minimizing the need for manual intervention.
- Given that T1-weighted MRI is a widely available clinical sequence, FD could be integrated into large-scale clinical imaging studies and patient stratification models with minimal workflow disruption.

Changes in manuscript:

We have made a small modification to emphasize the low computational cost of incorporating FD into routine clinical workflows. Otherwise, most of this language related to scalability/widespread adoption and accessibility was in our original discussion.

- Discussion: Lines 264-267

Reviewer #2 (Remarks on code availability):

Comment 12:

More commenting on the codes would be helpful for the reviewer to judge it.

Response:

We recognize the importance of well-documented code and have contacted the Code Ocean platform who have since helped us add more detailed commenting.

Changes in manuscript:

Our code now has additional commenting for the reviewers.

Reviewer #3 (Remarks to the Author):

This revised manuscript explores the prediction of medication changes in Parkinson's disease (PD) patients after deep brain stimulation (DBS) treatment based on the fractal dimension (FD) in T1-weighted magnetic resonance imaging (MRI) data. The revised manuscript used FD as a predictor of DBS treatment outcome in PD patients for the first time and validated the correlation between FD and PD disease severity. The revised manuscript used publicly available datasets from multiple centres for the validation of FD, enhancing the generalisability and reliability of the findings. The revised manuscript used both traditional clinical features (e.g., age, gender, DBS electrode location, etc.), and imaging features, such as FD, to construct a

multi-feature prediction model, and performed the classification learning task via a hypergraphic neural network (HGNN), which better captures the complex relationships between features to improve the performance of the prediction model.

Response:

We would like to clarify that this submission is the **original** version of our manuscript and has not been revised previously. We understand that the repeated use of "revised manuscript" throughout the review may have been unintentional, but we want to ensure there is no confusion regarding the submission history.

The revised version also has some problems:

Comment 1:

the description of subgraphs in the figure notes of Fig. 1 is not clear enough, perhaps it could be consistent with the figure notes of other figures.

Response:

The updated description provides a more detailed explanation of the subgraphs, aligning with the formatting and level of detail used in the rest of the manuscript.

Changes in manuscript:

The description for Figure 1 has been updated.

Comment 2:

the text of the tables in the revised version overlaps and is not legible.

Response:

We appreciate the reviewer bringing this issue to our attention. Upon reviewing the originally submitted table files, we confirmed that the formatting and legibility are correct in the separate table PDFs. However, it appears that the issue arose when the journal's submission system compiled the manuscript and figures into a single document for review, causing overlapping text artifacts. We will re-upload the tables' PDFs to minimize the likelihood of formatting issues in the next version.

Changes in manuscript:

We have re-uploaded the tables; however, we cannot guarantee that this compiling error will not appear again.

Comment 3:

the data used in the revised draft is from a different scanner and protocol, there may be data quality and consistency issues, how can this be resolved? Although the article argues that such data brings diversity and enhances the generalization of the model, it may also introduce noise, how did the authors distinguish and address this?

Response: To address potential concerns regarding data heterogeneity and scanner differences, we previously noted (Reviewer 2, comment 3) that while our dataset includes scans acquired on different systems, the acquisition parameters were relatively consistent across subjects, minimizing significant protocol-based variability. Although diversity in acquisition settings enhances model generalizability, we acknowledge that it is important to assess whether such variations introduce noise that could compromise reproducibility. To this end, we conducted an independent reproducibility analysis using a publicly available dataset acquired at

multiple time points. This analysis, detailed in the supplementary material, demonstrates strong short-term and long-term reproducibility of FD metrics, with high correlation coefficients ($R^2 > 0.92$) and over 94% of values falling within the limits of agreement in Bland-Altman tests. These results support the robustness of FD across acquisitions, reinforcing its suitability as a biomarker despite session-related variability.

Changes in manuscript:

- Methods: Lines 414-418
- Supplementary material: Lines 26-55

Comment 4:

Although the authors used multicentre data, the main analysis in the revised manuscript was based on 231 patients from a single centre. The sample size and diversity of the data may not be sufficiently representative of the broad population of PD patients, especially those with different disease duration and symptom severity.

Response:

This dataset was selected due to its high-quality imaging and detailed clinical outcome measures; however, we acknowledge the need for larger, more diverse cohorts to improve generalizability. Our ongoing work aims to expand the cohort by incorporating multi-site data to enhance external validity and account for heterogeneity in clinical presentation.

Changes in manuscript:

We have emphasized the need for multicenter data to validate our models in the main analysis and have invited readers to collaborate through data-sharing efforts.

- Discussion: Lines 297-303